# Impact of High-Intensity Interval Training on Body Composition and Depressive Symptoms in Adults under Home Confinement

**DOI:** 10.3390/ijerph19106145

**Published:** 2022-05-18

**Authors:** Diego Alonso-Fernández, Rosana Fernández-Rodríguez, Yaiza Taboada-Iglesias, Águeda Gutiérrez-Sánchez

**Affiliations:** 1Departament of Special Didactics, Faculty of Education and Sport Sciences, University of Vigo, 36005 Pontevedra, Spain; rosanafernandez@uvigo.es (R.F.-R.); yaitaboada@uvigo.es (Y.T.-I.); agyra@uvigo.es (Á.G.-S.); 2Education, Physical Activity and Health Research Group (Gies-10-DE3), Galicia Sur Health Research Institute (IIS Galicia Sur), SERGAS-UVIGO, 36208 Vigo, Spain

**Keywords:** COVID-19, pandemic, lockdown, HIIT, physical activity, depression

## Abstract

The home confinement derived from the COVID-19 pandemic has led to drastic changes in people’s habits. This situation has influenced their eating, rest, physical activity and socialization patterns, triggering changes in their mental stability. It was demonstrated that physical activity is beneficial for people’s physical and mental health. By its moderate volume and requiring little space or material, high-intensity interval training (HIIT) could prove to be a valid alternative in a situation of confinement. The aim of the present study was to observe the impact of an 8-week HIIT protocol on the body composition and the depressive symptoms of adults in strict home confinement. A total of 21 healthy adults, both male and female, (35.4 ± 5.6 years old; 70.50 ± 12.1 kg; 171 ± 10 cm) were divided into an experimental group (EG, n = 11) who carried out an 8-week Tabata protocol, based upon calisthenic exercises with their own weight in their homes, and a control group (CG, n = 10) who did not carry out any systematic physical activity over the same period. Following the intervention, the EG experienced a significant reduction in percentage (t = 3.86, d = 0.57, *p* < 0.05) and in kg (t = 4.62, d = 0.29, *p* < 0.05) of body fat mass (BFM) and body fat mass index (BFMI) (t = 4.61, d = 0.31, *p* < 0.05), as well as a reduction in depressive symptoms (t = 6.48, d = 1.3, *p* < 0.05). These results indicate that HIIT is a potential public health tool that could possibly be prescribed to the population in case of future situations of home confinement.

## 1. Introduction

In 2020, due to the worldwide COVID-19 pandemic, Spain was compelled to complete 7 weeks (from 14 March to 2 May) of strict home confinement, during which the population could only leave their homes in order to carry out justified tasks of utmost necessity. This period was characterized by fear and uncertainty, but also by a high impact on citizens’ lifestyle habits.

The pandemic disrupted people’s daily habits, raising the consumption of “comfort food” as a reaction to stress and boredom [1], diminishing the levels of physical activity [2] as well as increasing the rate of alcohol consumption [3]. Moreover, this situation also generated a higher incidence of psychiatric morbidity and psychological distress [4] generating anxiety and stress disorders linked to the pandemic, to the fear of contracting the disease, and to the widespread uncertainty concerning the future [5]. 

Scientific evidence is compelling in showing the beneficial influence that physical exercise has on the body composition of individuals [6]. Thus, training combining strength and resistance exercises induces a decrease in body fat mass and an increase in body lean mass in both sedentary men [7] and women [8]. In addition, regular physical exercise is associated with better mental health and lower stress levels regardless of age, gender, race or exercise type [9]. 

Unfortunately, the worldwide COVID-19 pandemic and the resulting confinement forced the population to confine themselves to their homes, the result being that regular physical exercise became more complicated due to lack of space and specific equipment.

High-intensity interval training (HIIT), which comprises repeated series of high-intensity effort followed by variable recovery times [10] has been shown to be an effective method of weight control and of reducing the total and visceral fat percentage in different population groups [11,12,13,14,15]. In addition, one of its main advantages over other lower-intensity training methods is that it triggers similar or even better results in a shorter session time [16]. For this reason, some authors have recently pointed out that HIIT could be the most effective and efficient way to improve health and reduce mortality in the adult population [17,18]. 

Given its characteristics, HIIT can be performed with little or no equipment and in a reduced space; it therefore appears to be, a priori, an ideal tool for maintaining the levels of physical exercise in a period of confinement. HIIT performed with body weight alone can promote significant adaptations in strength, hypertrophy [19], body composition and the cardiorespiratory system [13]. Moreover, HIIT-based interventions are tolerable and acceptable to previously sedentary individuals, generally showing lower dropout rates than those commonly reported for traditional exercise programs [18]. 

The present study is aimed at observing the impact of HIIT-based home training on body composition and depressive symptoms in an adult population with no previous experience, over a period of 8 weeks of strict confinement. 

## 2. Materials and Methods

### 2.1. Participants

The sample was comprised of 21 healthy adults, both male and female (10 women and 11 men) with ages ranging from 28 to 45, and a physically active lifestyle prior to the confinement period. The sample was randomly divided into two groups: an experimental group (EG, n = 11; 6 women and 5 men) and a control group (CG, n = 10; 4 women and 6 men) with no significant differences in their anthropometric characteristics prior to the intervention (Table 1).

### 2.2. Procedeures

The required inclusion criteria were not to have suffered any osteoarticular injury in the preceding 6 months, not to suffer from any medical condition which would prevent carrying out physical exercise and not having any previous experience with high-intensity interval training. All participants were duly informed of the features of the study and accepted to participate of their own volition, signing the required informed consent. The study was approved by the Autonomous Ethics Committee of Research of Xunta de Galicia, Ministry of Health (Spain) (registration no.: 2020/203) and all procedures were designed and administered in accordance with the Helsinki Declaration.

### 2.3. Research Design

The study was conducted over a period of 59 days, from 13 March to 10 May 2020. The first seven full weeks (16 March to 3 May) coincided with a situation of total home confinement for the participants, who were prevented from leaving their homes without a justified (work-related or medical) cause; the eighth week (4 May to 10 May) saw the beginning of the de-escalation, partially allowing citizens to leave their homes. Over 8 weeks, the EG carried out 2–3 daily HIIT sessions, based on functional exercises with their own weight (Alonso-Fernández et al., 2019) [13], synchronically taught and directed through a collective video conference (Zoom^®^), whilst the CG went about their daily lives in confinement without performing any systematic pattern of physical exercise.

Prior to the experimental phase, a session of anthropometric and body-composition measurements (M1, 13 March) was carried out at the participants’ homes using a Tanita RD-545 (Tanita Corp., Tokyo, Japan), with four electrodes and a “Holtex” tallimeter (Tanita Institute Contract Study, 2004) [20] which was then repeated at the end of said phase (M2, 10 May). The variables taken into account were weight (kg), body mass index (BMI) (kg/m^2^), fat mass (FM) (percentage and kg), fat body mass index (FBMI) (kg/m^2^), lean mass (LM) (kg) and the lean body mass index (LBMI) (kg/m^2^).

Likewise, participants’ depressive symptoms were recorded every 4 weeks (S1: 15 March; S4: 12 April; S8: 10 May) using the Spanish version [21] of Kandel and Davies’ six-item self-report scale [22] adjusted to the confinement situation. Thus, participants had to answer six questions related to social activities that may affect their health “over the past 4 weeks” using a scale ranging from 1 to 4 (“often”, “sometimes”, “rarely” or “never”). Previous studies have reported acceptable reliability of this scale [21] as well as that obtained in the present study (Cronbach’s alpha = 0.84 for the entire scale). Responses to the scale items were added up in order to obtain an overall depressive-symptom score ranging from 6 to 24 points. According to Choi et al. [23], the scores were increased by 10 points in order to obtain a range between 16 and 34 points, so that 29 points or more would be defined as marked depressive symptoms.

All participants were asked not to modify their eating habits during the intervention. Food intake was recorded before and after the intervention program by averaging three 24 h recalls conducted on nonconsecutive days (including one weekend day), which is a valid method for determining energy intake with an accuracy of 8–10% of actual energy intake [24]. In order to help estimate the amount of food consumed, color photographs of different food portion sizes were used. The NUTRIBER^®^ software (version 1.1.1) was used to determine the energy intake and macronutrient content derived from the three 24-h recalls.

#### HIIT Training

The EG’s intervention program, which used an HIIT protocol, was made up of high-intensity intermittent efforts based upon the “Tabata” method [25]. It consisted of a 4 min workout block with eight 20 s intensive workout intervals, each followed by 10 s of rest. The proposed exercises were structured in functional self-loading exercises, i.e., with own body weight and involving several joints and muscle groups, which had previously been used with positive results in the body composition and aerobic capacity of teenage subjects [13]. A warm up (based on a quick 5 min walk around the house and 10 min of joint mobility with dynamic stretching) was carried out on each occasion prior to the training.

The exercises and the development thereof were designed to be performed in a reduced space, in accordance with the dimensions of a private home. All participants were asked to perform the exercises at a high intensity, by increasing the execution speed during the 20 s of each workout interval. Intensity was monitored individually using heart-rate monitors and a subjective perception of effort scale.

All sessions were carried out simultaneously by the entire EG, remotely directed and supervised via group videoconference by the research team in order to guarantee their adequacy, homogeneity and intensity. The organization and order of the proposed block of exercises is shown in Figure 1.

The structure of the HIIT protocol intervention for the EG was distributed as follows during the 8 weeks of the experimental phase (Table 2):

### 2.4. Statistical Analysis

The analyses were performed using the SPSS software, version 25.0 for 195MacOS (IBM Corporation, Chicago, IL, USA). The data were revised in order to ensure the normality assumption by means of the Shapiro–Wilk test and homoscedasticity through Levene’s test. The Greenhouse–Geisser correction was used when the test of sphericity was violated (*p* < 0.05 for Mauchly’s test of sphericity). The changes in the dependent variables over time were compared between groups by a repeated-measures ANOVA (2 groups × 2 time points: pretest and posttest). A post hoc *t*-test was performed, along with the Bonferroni correction, to determine significant differences between pairwise comparisons. Significance was set at *p* < 0.05 and, where appropriate, Cohen’s d was used to establish the effect size, where values of 0.2, 0.5 and 0.8 represent small, medium and large differences, respectively [26]. 

## 3. Results

During the experimental phase, the calorie intake did not undergo significant changes in either of the groups (as shown in Table 3).

### 3.1. Body Composition

Table 4 shows the results obtained when examining the different body-composition variables measured in the sample of subjects before (M1) and after the intervention (M2).

Weight was not affected by the HIIT protocol, F (1.01–9.09) = 0.063, *p* > 0.05, $\eta_{p}^{2}$ = 0.07. The EG (t = 1.76, *p* > 0.05) did not show a significant variation in weight; however, in the CG a significant increase occurred (t = −3.65, d = 0.26, *p* < 0.05) after completion of the 8 weeks of the experimental phase. 

BMI was also unaffected by the HIIT protocol, F (1.05–9.43) = 0.27, *p* > 0.05, $\eta_{p}^{2}$ = 0.03. No significant changes were observed at the end of the 8-week experimental phase in the EG (t = 1.61, *p* > 0.05). By contrast, a significant increase in this variable was observed in the CG (t = −3.51, d = 0.52, *p* < 0.05).

The HIIT protocol did cause changes in the variables related to body fat mass. FM (%) was affected, F (1.04–9.3) = 3.06, *p* < 0.05, $\eta_{p}^{2}$ = 0.25. Regarding the pairwise comparisons derived from the post hoc analysis, the EG showed a significant decrease in the percentage of fat between M1 and M2 (t = 3.86, d = 0.57, *p* < 0.05) whereas the CG showed a significant increase in this variable (t = −7.67, d = 0.81, *p* < 0.001).

Likewise, FM (kg) varied as a result of the HIIT protocol, F (1.01–9.12) = 0.81, *p* < 0.05, $\eta_{p}^{2}$ = 0.82. In the pairwise comparisons, the EG showed a significant decrease in the kg of fat mass between M1 and M2 (t = 4.62, d = 0.29, *p* < 0.05) whereas the CG showed a significant increase in said variable (t = −6.62, d = 0.58, *p* < 0.05).

FMBI was also affected by the HIIT protocol, F (1.02–9.23) = 1.52, *p* < 0.05, $\eta_{p}^{2}$ = 0.34. The pairwise comparisons showed a significant decrease in FBMI between M1 and M2 in the EG (t = 4.61, d = 0.31, *p* < 0.05) and a significant increase in the CG (t = −7.25, d = 0.87, *p* < 0.001).

Nevertheless, LM (kg) was not affected by the HIIT protocol, F (1.02–9.19) = 0.41, *p* > 0.05, $\eta_{p}^{2}$ = 0.04. The pairwise comparisons showed no significant changes in this variable between M1 and M2 in either the EG (t = 4.62, *p* > 0.05) or the CG (t = −6.62, *p* > 0.05).

LBMI was likewise not affected by the HIIT protocol, F (1.13–10.14) = 2.16, *p* >0.05, $\eta_{p}^{2}$= 0.19, with no significant changes in this variable between M1 and M2 in either the EG (t = −0.67, *p* > 0.05) or the CG (t = −0.27, *p* > 0.05).

### 3.2. Depressive Symptoms

On the other hand, there were variations in the depressive symptoms as a result of the protocol, F (2.94–26.49) = 9.12, *p* <0.05, $\eta_{p}^{2}$ = 0.51. In the pairwise comparisons, there were no significant changes in this variable in the EG between weeks 1 and 4 (t = 3.31, *p* > 0.05), but they did appear between weeks 1 and 8 (t = 6.48, d = 1.3, *p* < 0.05). By contrast, no significant changes were observed in the CG in either the first four (t = 1.21, *p* > 0.05) or in the last four weeks of the experimental phase (t = 1.81, *p* > 0.05) in this variable (Figure 2).

## 4. Discussion

The COVID-19 pandemic caused situations of home confinement of varying intensity across the globe. These situations generated drastic changes in people’s lifestyle habits, limiting their mobility and social activities. Among the consequences reported were a reduction in physical activity levels and an increase in sedentary lifestyles [27,28], weight gain, poorer sleep quality [27], altered eating habits [27,29] and an adverse impact on mental health [30]. 

The potential of HIIT as a physical exercise alternative in confinement situations was theoretically pointed out, given its benefits for health [31], its positive effects on the immune system [32], reduced time commitment and minimal equipment and space requirements [32,33]. 

These theoretical potentialities have been tested in confinement situations to evaluate this type of exercise on stress, depression and resilience in healthy adults during confinement [34]. However, to the authors’ knowledge, this is the first study which could observe the impact of HIIT training based upon the Tabata method with self-loading exercises on the body composition as well as on the depressive symptoms of healthy adult individuals in a situation of strict confinement. 

The 8-week training period revealed positive consequences on the EG’s body composition with significant reductions in kg (−5.29%) and percentage of FM (−4.61%) and FBMI (−5.34%); these results are in line with those previously obtained by Amaro-Gahete et al. [14], with a 12-week HIIT intervention consisting of three weekly sessions in a group of adults. In addition, Soylu et al. [35] obtained a significant reduction in FM percentage (−16.2%) in a group of 14 healthy adults who performed a running-based HIIT protocol for 8 weeks. A recent meta-analysis, which included 39 studies, also concluded that HIIT significantly reduced body fat (−6%), especially visceral body fat (−8%), in sedentary individuals with BMI ranging from 18.5 to 35 kg/m^2^ [12]. These results back those obtained by a previous meta-analysis [36] in which HIIT also showed an impact on body-fat reduction (−7%). This fact seems to indicate that this type of exercise is an effective method for controlling the fat weight in confinement, even more so when one evaluates the results observed in the CG in the same variables, where there was a significant increase in weight (+2.66%), BMI (+2.81%), % (+8.43%) and kg of FM (+11.43%) and BMIF (+11.38%) although no significantly higher calorie intake had been observed. 

Regarding LM, the HIIT used did not have a significant impact on said variable. These results coincide with those previously obtained by Wewege et al. [36] and Nybo et al. [37] characterized by performing a HIIT protocol based on cyclical workout (running, cycling, and elliptical bike). However, these results are generally reversed when HIIT is based on self-loading calisthenic exercises, with an increase in the lean mass of the participants [13,14]. The present study implemented HIIT with calisthenic exercises based on body weight, and despite the fact that a slight increase in lean mass (0.42%) was indeed observed, this was not statistically significant. A possible explanation may be found in the participants’ lack of experience with HIIT, which may lead to a lower intensity in the execution or to the shorter experimental phase time (8 weeks) than in the case of previous studies [14]. Further research involving HIIT based on self-loading exercises would be necessary in order to ascertain whether this trend is consistent.

The depressive symptoms were significantly reduced (−20.3%) after 8 weeks in the group of subjects who performed the HIIT protocol. A recent review [38] has shown that the COVID-19 pandemic caused stress, anxiety, social isolation and psychological distress in adults, and levels of anxiety and depression higher than usual in frontline medical personnel. Physical activity is associated with increased wellbeing, improved cognitive functioning and decreased depression and anxiety [39]. Specifically, it reduces the severity, risk of onset and relapse of depression [40]. 

The results obtained for this variable are consistent with those obtained by the study also performed in the COVID-19 confinement by Borrega-Moquinho et al. [34] with a sample of 67 healthy adults. A total of 36 of them performed a HIIT protocol and 31 a MIT (moderate-intensity training) protocol of 8 weeks with self-loading exercises 6 days a week, obtaining a significant reduction in stress, anxiety and depression in both groups. However, according to these authors, the HIIT protocol was more beneficial in reducing depression than the MIT protocol. 

In a similar direction, the study by Méndez-Giménez et al. [39] observed that the subjects with a weekly frequency of 3–4 days of vigorous-intensity physical activity reduced noticeable depressive symptoms in confinement. 

These results are also consistent with pre-COVID-19 pandemic scientific evidence reporting the benefits of physical activity on mental health [41] and that its practice during leisure time, regardless of its intensity level, protects against future depression [41,42]. Thus, Arslan et al. [43] in a sample of premenopausal women obtained significant improvements in depressive symptoms after 8 weeks of aerobic training or aerobic training combined with strength exercises performed three days a week.

The authors are aware of the fact that this research had a series of limitations. The very conditions of the confinement situation made it impossible for any type of movement and social contact among the population to take place. Thus, it would have been very interesting to study a greater number of variables in the sample, such as muscle strength levels and aerobic capacity, and on a greater number of occasions during the experimental phase. However, the situation of general lockdown did not allow the researchers to avail themselves of valid tools, nor of the adequate context to proceed with these measurements. Undoubtedly, such contributions would enrich the results of the present study.

The main findings of the present study were that (1) HIIT demonstrated its viability as an exercise alternative in the home setting due to its reduced volume and low space and equipment requirements; (2) it had a positive impact on the body composition, as well as (3) on the depressive symptoms of the participating individuals.

## 5. Conclusions

Despite the limited resources and the externally imposed constraints of the study, we believe that HIIT demonstrated its potential to contribute to improving healthy lifestyle habits in situations of confinement, something that became quite relevant in the very special circumstances that we have all experienced since the beginning of the year 2020. Thus, in practical terms, we believe that this work may be the beginning of a line of research that should go deeper into the use of HIIT as a possible public health tool which could partially counteract the negative effects imposed on the population by a situation of home confinement. 

In short, HIIT training is a viable training strategy with a positive impact on body composition and reduction in depressive symptoms in a strict confinement situation. However, in future research it would be interesting to investigate how the levels of muscular strength and aerobic capacity respond to HIIT training in confinement.

## Figures and Tables

**Figure 1 ijerph-19-06145-f001:**
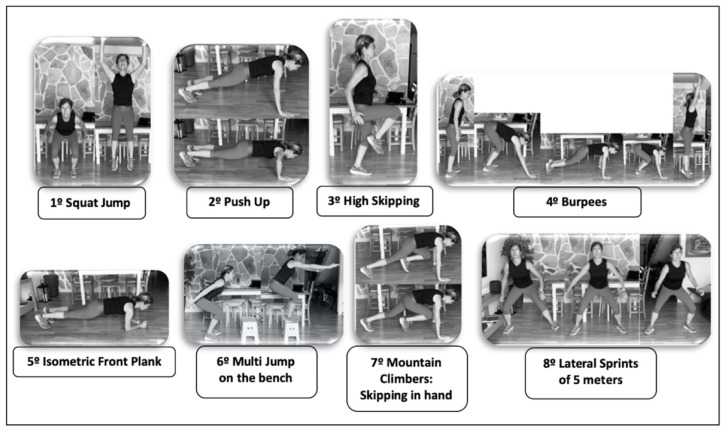
Organization of HIIT training exercises in the EG (4-min Tabata Block).

**Figure 2 ijerph-19-06145-f002:**
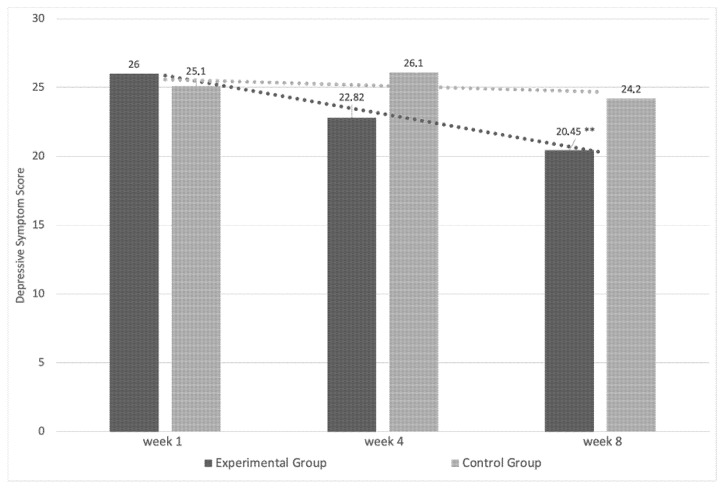
Measurement of depressive symptoms for EG and CG during experimental phase. ** = *p* < 0.05 vs. week 1.

**Table 1 ijerph-19-06145-t001:** Characteristics of the participants (mean ± *SD*).

Group	N	Age (Years Old)		Weight (kg)		Height (cm)	
		$\left(\bar{x}\right)$	(SD)	$\left(\bar{x}\right)$	(SD)	$\left(\bar{x}\right)$	$\left(S\bar{D}\right)$
EG	11	34.4	4.9	72.56	12.26	170	10
CG	10	36.5	6.4	70.56	11.02	172	11

CG: Control Group; EG: Experimental Group.

**Table 2 ijerph-19-06145-t002:** Timing and loading of the HIIT method for EG.

	“Tabata” Blocks	Min. Break between Blocks	Nº Exercises/Session	Nº Sessions per Week	HIIT Minutes per Week
Weeks 1 and 2	2	2.5	16	2	16
Week 3	3	2	24	2	24
Weeks 4 and 5	3	2	24	3	36
Weeks 6 and 7	3	1	24	3	36
Week 8	4	2	32	3	48

EG: Experimental Group; HIIT: High-Intensity Interval Training; Nº: Number.

**Table 3 ijerph-19-06145-t003:** Changes in diet in the EG and the CG between M1 and M2 (mean ± *SD*).

	EG (n = 11)			CG (n = 10)		
Variables	M1	M2	% Change	M1	M2	% Change
Kcal/day	2149.1 ± 413	2203.3 ± 511	2.51	2228.8 ± 488	2335.6 ± 402	4.81
Proteins/day (%)	18 ± 4	17 ± 3	−1	17 ± 2	17 ± 3	0
Lipids/day (%)	39 ± 5	41 ± 6	2	40 ± 7	43 ± 5	3
Carbohydrates/day (%)	43 ± 7	42 ± 5	−1	43 ± 8	40 ± 7	−3

CG: Control Group; EG: Experimental Group; M1: Measure 1; M2: Measure 2.

**Table 4 ijerph-19-06145-t004:** Changes in body composition in the EG and CG between M1 and M2 (mean ± *SD*).

	GE (n = 11)			GC (n = 10)		
Variables	M1	M2	% Change	M1	M2	% Change
Weight (kg)	72.56 ± 12.26	71.84 ± 11.77	−0.99	70.56 ± 11.02	72.44 ± 11.53	2.66 *
BMI (kg/m^2^)	24.54 ± 2.74	24.32 ± 2.68	−0.89	23.84 ± 1.86	24.51 ± 2.26	2.81 *
FM (%)	24.09 ± 4.31	22.98 ± 3.82	−4.61 *	25.20 ± 3.98	27.31 ± 4.21	8.37 **
FM (kg)	17.75 ± 5.16	16.81 ± 4.73	−5.29 *	18.01 ± 5.02	20.07 ± 5.71	11.43 *
FBMI (kg/m^2^)	5.99 ± 1.59	5.67 ± 1.43	−5.34 *	6.06 ± 1.34	6.75 ± 1.54	11.38 **
LM (kg)	54.81 ± 7.94	55.04 ± 7.63	0.42	52.55 ± 6.85	52.37 ± 6.61	−0.34
LBMI (kg/m^2^)	18.56 ± 1.35	18.66 ± 1.42	0.54	17.78 ± 0.85	17.75 ± 1.09	−0.17

* = *p* < 0.05 vs. M1: ** = *p* < 0.001 vs. M1: Measure 1; M2: Measure 2; CG: Control Group; EG: Experimental Group; M1: Measure 1; M2: Measure 2.; BMI: body mass index; FM: fat mass; FBMI: fat body mass index; LM: lean mass; LBMI: lean body mass index.

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
