# Peer review of "Impact of High-Intensity Interval Training on Body Composition and Depressive Symptoms in Adults under Home Confinement"

_ijerph, 2022, doi:10.3390/ijerph19106145_

Round 1

Reviewer 1 Report

General

This article dealt with a very interesting and current topic.
The authors aimed to observe the impact of an 8-week HIIT protocol on the body composition and the depressive symptoms of adults in strict home confinement over the pandemic period. The results showed a positive impact on body composition and depressive symptoms.

The article is well written. I have a few suggestions for improving the manuscript.

Specific comments

Keywords

Please, remove the keywords already present in the title (confinement, body composition) with other different ones, if necessary. To optimize the search for the manuscript through search engines, it is good to use keywords other than the title.

Material and Methods

Line 86: In my opinion this paragraph should be called "Procedures".

Results

If possible, divide the Results into subsections to make reading easier, for example, using the two measured dependent variables: Body Composition and Depressive Symptoms.

Line 163: Replace with "... and M2 (mean ± SD)".

Table 4: Please, insert notes with the meaning of all the acronyms used: BMI, FM, FBMI, LM, LBMI

In Figure 2, replace "GE" and "GC" with "EG" and "CG"

Discussion

Line 207: "mental health" is generic; it does not represent the measured dependent variable. Replace with "Depressive symptoms".

Clearly describe the strengths of the study (if possible, write them after study limitations)

Conclusions

Please, put the limitations in a paragraph in the discussions (last).

In the conclusion you have to put the take-home message and the future implications.

Author Response

General

This article dealt with a very interesting and current topic.
The authors aimed to observe the impact of an 8-week HIIT protocol on the body composition and the depressive symptoms of adults in strict home confinement over the pandemic period. The results showed a positive impact on body composition and depressive symptoms.

The article is well written. I have a few suggestions for improving the manuscript.

First of all, let us thank you for your advice and recommendations. Below we detail the changes made following your indications. We believe that with your help we have been able to improve the document, thank you very much.

Specific comments

Keywords

Please, remove the keywords already present in the title (confinement, body composition) with other different ones, if necessary. To optimize the search for the manuscript through search engines, it is good to use keywords other than the title. Ok, Deleted keywords. Line 27

Material and Methods

Line 86: In my opinion this paragraph should be called "Procedures". Ok, paragraph added as "Procedures". Line 80

Results

If possible, divide the Results into subsections to make reading easier, for example, using the two measured dependent variables: Body Composition and Depressive Symptoms. Ok, results divided into sub-sections. Line 172, 207

Line 163: Replace with "... and M2 (mean ± SD)". OK, replaced. Line 170

Table 4: Please, insert notes with the meaning of all the acronyms used: BMI, FM, FBMI, LM, LBMI. OK, inserted notes. Table 4.

In Figure 2, replace "GE" and "GC" with "EG" and "CG". OK, replaced. Figure 2.

Discussion

Line 207: "mental health" is generic; it does not represent the measured dependent variable. Replace with "Depressive symptoms". OK, replaced. Line 342

Clearly describe the strengths of the study (if possible, write them after study limitations). Ok, restructured the "discussion" and "conclusions" sections. Line 218, 344

Conclusions

Please, put the limitations in a paragraph in the discussions (last). Ok, "limitations" added to the discussion. Line 331

In the conclusion you have to put the take-home message and the future implications. Ok, added. Line 353

Reviewer 2 Report

I just have a few minor comments:

  1. In the titles of tables 1 and 3 there is an error in the name of the SD.
  2. In Table 1, body height is given in cm, not in m.
  3. There are no significant differences in Table 3, so it is not necessary to explain what * means.
  4. In table 4, please recalculate the% change for FM% in the GC group.
  5. There is no information in Figure 2 about the variables on the x and y axes. As a rule, the graph should read itself.

Author Response

First of all, let us thank you for your advice and recommendations. Below we detail the changes made following your indications. We believe that with your help we have been able to improve the document, thank you very much.

I just have a few minor comments:

  1. In the titles of tables 1 and 3 there is an error in the name of the SD. OK, replaced. Table 1, 3
  2. In Table 1, body height is given in cm, not in m. OK, replaced. Table 1
  3. There are no significant differences in Table 3, so it is not necessary to explain what * means. OK, deleted. Table 3
  4. In table 4, please recalculate the% change for FM% in the GC group. OK, recalculated. Table 4
  5. There is no information in Figure 2 about the variables on the x and y axes. As a rule, the graph should read itself. OK, modified graph. Figure 2

Reviewer 3 Report

Impact of high-intensity-interval training on body composition and depressive symptoms in adults under home confinement

First of all, the reviewer would like to thank the authors for their work and efforts in trying to improve sports science knowledge.

General comments to the authors

Overall, this is a nice study that could have great observe the impact of an 8-week HIIT protocol 16 on the body composition and the depressive symptoms of adults in strict home confinement. The authors are commended on their efforts thus far. The study is well designed and well-written, with a great original article evaluating the usefulness of the topic. However, I suggest only small corrections and the authors should update the recent references about the topic, these corrections and studies will allow improving the manuscript.

Abstract

This section is well designed and well-written.

Introduction section

This section is well designed and well-written.

Methods section

This section is well designed and well-written.

Results section

Please be careful to add units for the depressive symptoms in figure 2. Instead of this, bar graph is better than this

Discussion section

Overall the discussion is well-written and incorporates relevant literature. The authors should add these references in discussion to support their ideas

Borrega-Mouquinho, Y., Sánchez-Gómez, J., Fuentes-García, J. P., Collado-Mateo, D., & Villafaina, S. (2021). Effects of high-intensity interval training and moderate-intensity training on stress, depression, anxiety, and resilience in healthy adults during coronavirus disease 2019 confinement: a randomized controlled trial. Frontiers in Psychology, 12, 270.

Arslan, E., Can, S., & Demirkan, E. (2017). Effect of short-term aerobic and combined training program on body composition, lipids profile and psychological health in premenopausal women. Science & Sports, 32(2), 106-113.

de Paula, C. C., Machado, S., Costa, G. D. C. T., Sales, M. M., Miranda, T. G., Barsanulfo, S. R., ... & Sá Filho, A. S. (2020). High intensity interval training (HIIT) as a viable alternative to induce the prevention of respiratory diseases: a point of view of exercise immunology during COVID-19 outbreak. Research, Society and Development, 9(10), e7069109186-e7069109186.

Soylu, Y., Arslan, E., Sogut, M., Kilit, B., & Clemente, F. M. (2021). Effects of self-paced high-intensity interval training and moderate-intensity continuous training on the physical performance and psychophysiological responses in recreationally active young adults. Biology of Sport, 38(4), 555.

Müller, C. B., Veiga, R. S. D., Pinheiro, E. D. S., & Vecchio, F. B. D. (2021). Home-based high-intensity interval training can improve physical performance in young female athletes during a quarantine. Motriz: Revista de Educação Física, 28.

Fuentes-García, J. P. (2020). Impact of high intensity interval training and aerobic training on stress, depression symptoms, anxiety and resilience in self-isolated people during the COVID-19 health emergency in Spain.

Figures and Tables

This section is well designed and well-shown.

Author Response

First of all, the reviewer would like to thank the authors for their work and efforts in trying to improve sports science knowledge.

First of all, let us thank you for your advice and recommendations. Below we detail the changes made following your indications. We believe that with your help we have been able to improve the document, thank you very much.

General comments to the authors

Overall, this is a nice study that could have great observe the impact of an 8-week HIIT protocol 16 on the body composition and the depressive symptoms of adults in strict home confinement. The authors are commended on their efforts thus far. The study is well designed and well-written, with a great original article evaluating the usefulness of the topic. However, I suggest only small corrections and the authors should update the recent references about the topic, these corrections and studies will allow improving the manuscript.

Abstract

This section is well designed and well-written. Ok, thanks a lot

Introduction section

This section is well designed and well-written. Ok, thanks a lot

Methods section

This section is well designed and well-written. Ok, thanks a lot

Results section

Please be careful to add units for the depressive symptoms in figure 2. Instead of this, bar graph is better than this. Ok, modified graph. Figure 2

Discussion section

Overall the discussion is well-written and incorporates relevant literature. The authors should add these references in discussion to support their ideas

Thank you very much for the papers you have indicated. We believe that they have contributed to improve the discussion of our document. 

Borrega-Mouquinho, Y., Sánchez-Gómez, J., Fuentes-García, J. P., Collado-Mateo, D., & Villafaina, S. (2021). Effects of high-intensity interval training and moderate-intensity training on stress, depression, anxiety, and resilience in healthy adults during coronavirus disease 2019 confinement: a randomized controlled trial. Frontiers in Psychology12, 270. OK, added. Line 242, 281

 Arslan, E., Can, S., & Demirkan, E. (2017). Effect of short-term aerobic and combined training program on body composition, lipids profile and psychological health in premenopausal women. Science & Sports32(2), 106-113. OK, added. Line 293

de Paula, C. C., Machado, S., Costa, G. D. C. T., Sales, M. M., Miranda, T. G., Barsanulfo, S. R., ... & Sá Filho, A. S. (2020). High intensity interval training (HIIT) as a viable alternative to induce the prevention of respiratory diseases: a point of view of exercise immunology during COVID-19 outbreak. Research, Society and Development9(10), e7069109186-e7069109186. OK, added. Line 226

 Soylu, Y., Arslan, E., Sogut, M., Kilit, B., & Clemente, F. M. (2021). Effects of self-paced high-intensity interval training and moderate-intensity continuous training on the physical performance and psychophysiological responses in recreationally active young adults. Biology of Sport38(4), 555. OK, added. Line 250

 Müller, C. B., Veiga, R. S. D., Pinheiro, E. D. S., & Vecchio, F. B. D. (2021). Home-based high-intensity interval training can improve physical performance in young female athletes during a quarantine. Motriz: Revista de Educação Física28. We have carefully reviewed the document. But we believe that this is based on a very short intervention (2 weeks of training) and with variables different from ours (Perception of enjoyment and number of repetitions in each exercise performed), it would not be advisable to include it in our discussion.

 Fuentes-García, J. P. (2020). Impact of high intensity interval training and aerobic training on stress, depression symptoms, anxiety and resilience in self-isolated people during the COVID-19 health emergency in Spain. This work is the preliminary version of the one you indicated above: Borrega-Mouquinho et al., (2021)

 Figures and Tables

This section is well designed and well-shown. Ok, thanks a lot